# Multitask Learning-Based Deep Signal Identification for Advanced Spectrum Sensing

**DOI:** 10.3390/s23249806

**Published:** 2023-12-13

**Authors:** Hanjin Kim, Young-Jin Kim, Won-Tae Kim

**Affiliations:** 1Future Convergence Engineering Major, Department of Computer Science and Engineering, Korea University of Technology and Education, Cheonan 31253, Republic of Korea; gks359@koreatech.ac.kr; 2Department of Computer Science Engineering, Korea University of Technology and Education, Cheonan 31253, Republic of Korea; you359@koreatech.ac.kr

**Keywords:** deep signal identification, multitask learning, spectrum sensing, spectrum hyperspace, 5G-advanced

## Abstract

The explosive demand for wireless communications has intensified the complexity of spectrum dynamics, particularly within unlicensed bands. To promote efficient spectrum utilization and minimize interference during communication, spectrum sensing needs to evolve to a stage capable of detecting multidimensional spectrum states. Signal identification, which identifies each device’s signal source, is a potent method for deriving the spectrum usage characteristics of wireless devices. However, most existing signal identification methods mainly focus on signal classification or modulation classification, thus offering limited spectrum information. In this paper, we propose DSINet, a multitask learning-based deep signal identification network for advanced spectrum sensing systems. DSINet addresses the deep signal identification problem, which involves not only classifying signals but also deriving the spectrum usage characteristics of signals across various spectrum dimensions, including time, frequency, power, and code. Comparative analyses reveal that DSINet outperforms existing shallow signal identification models, with performance improvements of 3.3% for signal classification, 3.3% for hall detection, and 5.7% for modulation classification. In addition, DSINet solves four different tasks with a 65.5% smaller model size and 230% improved computational performance compared to single-task learning model sets, providing meaningful results in terms of practical use.

## 1. Introduction

The massive connectivity of wireless devices has caused a significant increase in spectrum demand, leading to a shortage in the available spectrum. To address this, wireless devices tend to (1) share the spectrum for opportunistic spectrum utilization and (2) apply various wireless technologies to densely use multidimensional spectrum space. The explosive demand for the spectrum is expected to increase further, even in 5G-advanced and 6G systems, and the network must accommodate heterogeneous devices and their complex access methods [1,2,3].

Spectrum sensing (SS), serving as the cornerstone of cognitive radio, has played a crucial role in mitigating spectrum scarcity by identifying unused resources within the spectrum and promoting their use by wireless devices [4]. Nonetheless, as spectrum dynamics become increasingly complex, wireless devices are demanding a more advanced level of detection from SS. To enhance spectrum efficiency with minimal interference amidst frequent spectrum sharing, it is essential for SS to not just detect ’empty holes’ but also keep track of the activities of primary users [5]. Furthermore, to mitigate cross-technology interference (CTI) among devices utilizing diverse wireless technologies, there is a need to extract multidimensional spectrum state information, which is key in developing tailored interference mitigation strategies [6]. Given the functional limitations of SS, if wireless devices are confined to receiving fragmented information about the spectrum state, they may face frequent collisions and transmission delays [7]. Hence, it becomes critical for SS to evolve in a manner that enables the derivation of comprehensive spectrum state information, including time, frequency, power, and other relevant dimensions [8].

Signal identification [9] has the potential to be one of the key technologies facilitating the SS into advanced SS systems. It works by identifying signals from received radio frequency (RF) samples, enabling the SS system to infer spectrum usage characteristics of the identified signal, such as the channel in use, signal strength level, and modulation method. Numerous approaches have been explored for signal identification, including the use of cyclostationarity [10], maximum likelihood estimation [11], and the received signal strength indicator (RSSI) [12]. Recently, studies have focused on using machine learning or deep learning models, which are capable of capturing the characteristics of a signal source, and can replace static filters or statistical models that may not operate well in dynamic communication environments [13,14,15].

However, most signal identification studies conducted so far tend to align more with signal classification or modulation classification rather than the ’identification’, which is defined as recognizing something and verifying its characteristics [16]. In other words, it is challenging to obtain comprehensive spectrum state information based only on the limited information about the presence of a specific signal within the spectrum. Of course, knowing the existence of certain signals allows us to estimate their spectrum usage characteristics, but it is difficult to derive their characteristics at the detected point. While the signal footprint left by signals using the multidimensional spectrum space, spectrum hyperspace [8], can be used for signal classification, leveraging it to derive used characteristics in each spectrum dimension can be seen as a more beneficial strategy. For clarity, we distinguish between deep signal identification, which adheres to the aforementioned definition, and shallow signal identification, which identifies only a single characteristic, such as signal classification or modulation classification.

The deep signal identification problem we are interested in is quite challenging, as it requires simultaneous extraction of not only the signal’s classification but also its multidimensional usage characteristics. Considering the high adaptability of deep learning models, one might imagine creating multiple models for each individual identification task and carrying out deep signal identification. However, it is not that simple. As the number of signal properties to be identified increases, additional individual models need to be used, which might be difficult to accommodate in communication systems with limited resources, including access points and base stations [17]. Moreover, models biased towards individual tasks can overlook the correlation between signal footprints from the same signal source, making it difficult to capture multidimensional characteristics.

We posit that multitask learning (MTL), with its proven efficacy in computer vision [18], speech recognition [19], and natural language processing [20], is a prime solution to these challenges. Given MTL’s capability to sufficiently learn a diverse array of characteristics from the signal source via a single model, it can yield substantial signal identification results. In this paper, we present DSINet, a multitask learning-based deep signal identification model. DSINet serves as a key component of an advanced spectrum sensing system for forthcoming 5G-advanced and 6G networks. The main contributions of this paper can be summarized as follows.

We introduce an advanced spectrum sensing system designed to derive the multidimensional spectrum usage characteristics of wireless devices. The extraction process is formulated as a deep signal identification problem where each task derives usage characteristics for individual dimensions, such as time, frequency, power, and code.We propose DSINet, a multitask learning model explicitly tailored to address the deep signal identification problem. This model operates under the assumption that distinct wireless devices leave identifiable ’footprints’ within a multidimensional spectrum space. DSINet captures these ’footprints’ as common features that directly contribute to the resolution of individual tasks.The DSINet is designed as a modular architecture, composed of three distinct types of blocks: hard sharing blocks, soft sharing blocks, and task-specific blocks. Hard sharing blocks and soft sharing blocks concentrate on common and task-dependent features, respectively, fostering effective information sharing. Conversely, task-specific blocks target unique features inherent to each task. The integration of these blocks enhances overall signal identification performance.We conduct three case studies to evaluate DSINet. Each study not only involves a comparative evaluation with existing shallow signal identification technology, but also includes a performance investigation based on task correlation and an evaluation of model size and computational efficiency. The comprehensive results of these studies emphasize DSINet’s potential and practical utility for solving deep signal identification problems.

## 2. Background and Motivation

### 2.1. Background

The primary focus of SS is to detect opportunities for the unused spectrum that primary users (PUs) are not using in real time. To detect these gaps, various techniques, such as filter-based [21] and statistical model-based [22] methods, have been studied. However, given the inadequacies of static filters and statistical models in fully capturing actual communication environments, there has been a shift towards machine learning- and deep learning-based cognitive radio (CR) technologies [23,24]. These technologies adeptly extract unique characteristics of different communication environments, leading to significant performance improvements in terms of detection accuracy.

Recent studies have accorded considerable attention to estimating PU activity statistics and occupancy patterns, given their substantial impact on detection performance. For instance, one study [5] improved spectrum accuracy by detecting PU activity statistics in the time domain. Another study [25] enhanced the precision of channel occupancy estimation for wideband SS by considering the correlation between spectrum occupancy levels in adjacent channels. Given the robust performance of these studies, SS has evolved from mere hole detection [2] to a more nuanced concept that quantifies spectrum usage statistics and identifies signal interference across frequency bands at a specific location. As spectrum sharing expands into multidimensional spectral spaces, these new perspectives on spectral sensing will gain further importance.

SS is also highly important in unlicensed bands. With the growing popularity of unlicensed bands, wireless technologies such as WiFi, ZigBee, and Bluetooth are vying for spectrum access, thereby leading to CTI [26]. To attenuate interference among wireless devices in the spectrum, it is essential to consider the varying levels of interference among them and apply appropriate mitigation techniques in each situation. However, energy detectors [27] and simple filter-based detectors [28,29], which are extensively used in unlicensed bands, furnish only limited spectrum state information, proving inadequate for the application of appropriate mitigation methods [7].

In the impending 5G-advanced and 6G environments, scenarios of simultaneous spectrum sharing and CTI are expected to be prevalent. The role of SS is set to become increasingly important in these transitions. In essence, for more efficient utilization of the spectrum, it is crucial that SS evolves to adeptly extract the multidimensional spectrum usage characteristics of each signal within the spectrum.

### 2.2. Motivation

#### 2.2.1. Deep Signal Identification

Signal identification is one of the technologies that can advance SS to function properly in complex spectrum dynamics. In [9], authors stated that understanding the use of spectrum resources to improve spectrum utilization would be an important asset for 5G, with the identification of radio signals at the core. Ali Gorcin [8] articulates that effective use of spectrum hyperspace dimensions demands more than just deciding the presence of primary users in the channel. It also necessitates the identification of primary user waveforms, such as air interface parameters, burst structures, chip rates, cyclic prefix sizes, preambles, and the radio access methods used, all without the need for demodulation.

Traditionally, signal identification has been explored through methods grounded in statistical and probabilistic approaches, such as cyclostationarity [10] and likelihood [11]. However, due to the limitations of these conventional techniques, including low accuracy and high computational complexity, research on feature-based signal identification methods has become more prevalent. The central premise of these methods revolves around utilizing the unique ’signal footprints’ left in the spectrum during communication. These are treated as valuable features for discerning the corresponding signals. Some deep learning models are employed to comprehend these signal characteristics, including the type of modulation used and the frequency band occupied. Research evidence demonstrating the precise identification of target signals in complex communication environments through learning these features shows the superiority of feature-based methods over traditional approaches [13,14,15].

Despite improvements in the accuracy of signal identification technologies, the realization of advanced SS requires progress from different perspectives. The term ’Identification’ refers to recognizing something and validation its properties [11], but in a strict sense, most studies fall under signal classification [17] and modulation classification [16]. Classified signal types or modulations can indicate the presence of specific signals in the current spectrum, but they cannot infer the spectrum usage characteristics of each signal. In other words, most research only performs shallow signal identification, which is insufficient information for executing spectrum sharing and CTI migration in complex spectrum dynamics.

As an alternative, the construction of multiple independent shallow identification models to derive multidimensional usage characteristics could be considered, but this introduces new problems, including increased memory size, processing speed, and system complexity. If shallow signal identification models are established for each dimension to extract the multidimensional spectrum usage characteristics required in advanced SS, the overall processing speed would increase linearly with each dimension to be detected, which is undesirable. Moreover, if separate preprocessing modules are additionally needed for each model, the complexity of chip configuration could significantly increase.

In the context of advanced SS, the importance of deep signal identification is prominent. The primary challenge resides in overcoming the limitations of multiple model-based approaches.

#### 2.2.2. Multitask Learning

Traditionally, neural networks have been primarily focused on processing individual tasks. However, most of the problems around us are essentially multitasks, such as self-driving cars [30], intelligent advertising systems [31], etc. MTL is a learning paradigm [32] that aims to leverage useful information contained in multiple related tasks to help improve the generalization performance of all the tasks. It has shown remarkable performance compared to single-task learning in computer vision, speech recognition, and natural language processing. MTL offers three primary benefits: reduced memory footprint, increased inference speed, and most importantly, performance enhancement via complementary information and mutual regularization of each task [33].

Signal identification for SS is also a type of multitasking, as it involves recognizing signals and verifying their multiple attributes. The multidimensional spectrum utilization characteristics of a signal could be considered a collection of tasks, each deriving signal usage characteristics corresponding to individual dimensions, such as time, frequency, and power. A proper configuration of multitasks for the dimensions to be identified, coupled with the sophisticated design of an MTL model, can achieve ideal deep signal identification.

While the application of MTL for deep signal identification can be promising, it necessitates careful consideration before implementation. Challenges that need addressing include overall performance degradation due to an improperly configured multitasking setup, task interference due to low correlations, and dependencies on specific tasks at different levels of task difficulty [31]. In advanced SS scenarios where deep signal identification is required, there can be conflicting attributes among identification tasks, and an improper application of MTL may lead to undesirable results. Please see Section 4 for more details. In this paper, we aim to explore in depth the potential of MTL-based deep signal identification, taking into account the interrelations among signal identification tasks, through various case studies and experiments.

## 3. Deep Signal Identification in Advanced Spectrum Sensing Systems

### 3.1. Concept of Advanced Spectrum Sensing System

Figure 1 illustrates the operation of an advanced SS system. *N* wireless devices in the same spectrum band use *W* wireless technologies to communicate with each other. Some devices are secondary users performing spectrum sharing, which is considered one of the wireless technologies. Signals with similar characteristics can be grouped to reduce the number of types that need to be identified [34].

The system comprises a down converter that lowers the signal band to the baseband, an analog–digital converter that digitizes analog signals, and a deep signal identification module that provides the spectrum usage characteristics of each signal to be sensed within the spectrum. We use the raw I/Q sample set to avoid system complexity instead of adding a preprocessing module, such as fast Fourier transform (FFT) or short-time Fourier transform (STFT), to obtain the transformation value.

As wireless devices communicate, they leave different footprints in the spectrum hyperspace, including time, frequency, and power footprints, as well as others [8]. This is formulated as follows: (1)$$S\leftarrow \sum_{i=1}^{N}h_{i}\left(g_{i}\left(s_{i}\right)\right).\;$$
where S is the spectrum hyperspace, *s* is the signal source of a wireless device, g is a function of the wireless technology used by the wireless device, and h is the wireless channel of each device. Since wireless devices can use the same wireless technology, *N* and *W* are not the same.

The goal of advanced SS is to derive the spectrum usage characteristics of wireless devices from S. Specifically, it aims to identify a set of properties for signal sources that use a specific wireless technology within the spectrum band. By sensing the target band, advanced SS can obtain sample *x* and derive the spectrum usage characteristics through these samples, which can be expressed as follows:(2)$$\hat{Y}=F\left(x\right),\forall x\in S.\;$$
where $\hat{Y}={\left\{{\hat{y}}_{p}\right\}}_{p\in P}$ is the set of P pieces of spectrum state information to be estimated; *x* denotes the received RF signal samples, i.e., I/Q pairs; and $F={\left\{f_{p}\right\}}_{p\in P}$ is a set of functions for estimating the P pieces of spectrum state information in SS.

### 3.2. Deep Signal Identification with Multitask Learning

Deep signal identification refers to the set of functions F that estimate $\hat{Y}$. One possible approach for estimating $\hat{Y}$ using F is to use individual machine learning model sets (independent shallow signal identification models), as shown below:(3)Y^=y^1=f1(x;θ1)y^i=⋮fi(x;θi)y^p=fp(x;θp)
where θ is the weight parameter for each corresponding function f.

However, this approach is not optimal in terms of its system complexity, development costs, and dependence on a preprocessing module. Our proposed multitask learning-based deep signal identification approach derives $\hat{Y}$ from the same sample of *x* as that used by a single MTL model:(4)$$\hat{Y}=f_{MTL}\left(x;\theta_{MTL}\right)\;$$

This approach not only overcomes the limitations of individual model sets but also learns the correlations of each task.

### 3.3. MTL-Based Deep Signal Identification Network

#### 3.3.1. Design Principles


**Principle 1. Hybrid Shared Network Structure**


MTL models have more weight parameters to be tuned during training than single-task learning models [35]. To establish effective learning strategies, we consider a hybrid network structure that combines hard sharing blocks (HBs) and soft sharing blocks (SBs). HBs extract common features shared by tasks through tightly coupled sharing layers, while an SB has a different layer for each task and shares information through a unit called a cross-stitch [36]. An HB uses unified shared layers, resulting in less memory usage and faster inference, but this may cause competition among tasks. In contrast, an SB consists of individual layers for each task, including the shared part, resulting in relatively less competition for each task.


**Principle 2. Task-Specific Network Structure**


In MTL, certain tasks may have different levels of difficulty or exhibit opposite characteristics compared to those of other tasks. These gaps and conflicts between tasks can impede the entire learning process, including specific tasks. To address this issue, we employ task-specific blocks (TBs).

#### 3.3.2. Joint Optimization of Multiple Tasks

In multitask learning, MTL model training is performed by minimizing the losses of all tasks. In other words, the joint optimization of all tasks is completed as follows:(5)$$L_{MTL}=\frac{1}{M}\sum_{t=1}^{M}\beta_{t}L_{t}\;$$
where *M* is the number of tasks, $L_{t}$ is the single loss associated with task *t*, and $\beta_{t}$ is the weight for loss $L_{t}$.

As we consider heterogeneous MTL, which consists of different types of supervised tasks, including classification and regression problems, the loss function for each individual task is one of the binary cross-entropy loss ($L_{BCE}$), the cross-entropy loss ($L_{CE}$), and the mean squared error loss ($L_{MSE}$).

The binary cross-entropy is formulated as follows:(6)$$L_{BCE}=\frac{1}{K}\sum_{i=1}^{K}y^{\left(i\right)}log\left({\hat{y}}^{\left(i\right)}\right)+\left(1-y^{\left(i\right)}\right)log\left(1-{\hat{y}}^{\left(i\right)}\right)\;$$
where *K* represents the number of binary labels, and *y* and $\hat{y}$ are the vectors of ground truth and predicted probabilities, respectively; each vector consists of *K* elements with values between 0 and 1. When *K* is 1, it is used for binary classification tasks, such as determining the presence of a specific signal or the use of a single channel. When K>1, the function is used for multilabel classification.

The $L_{CE}$ is defined as
(7)$$L_{CE}=-\frac{1}{C}\sum_{i=1}^{C}y^{\left(i\right)}log\left({\hat{y}}^{\left(i\right)}\right)\;$$

The cross-entropy loss is used for multiclass classification tasks, where *C* represents the number of classes. In this case, *y* denotes the ground truth vector in a one-hot encoded format, and $\hat{y}$ is the predicted probability distribution across *C* classes.

Due to the presence of complex spectrum states, advanced SS frequently encounters heterogeneous MTL problems in which both classification and regression tasks must be solved simultaneously. The loss function used for the regression task is as follows:(8)$$L_{MSE}=-\frac{1}{R}\sum_{i=1}^{R}{\left(y^{\left(i\right)}-{\hat{y}}^{\left(i\right)}\right)}^{2}\;$$
where *R* represents the number of regression targets, and *y* and $\hat{y}$ are the vectors of the ground truth target and the model’s prediction, respectively; each vector consists of *R* elements with continuous values. When *R* is 1, it is used for a single regression, and when R>1, it is used for multiple regression.

#### 3.3.3. Model Architecture

A set of tasks to be solved by DSINet is $T={\left\{T_{1},T_{2},...,T_{i}\right\}}_{i=1}^{M}$, which can include signal classification, modulation classification, channel classification, and power estimation. The number of tasks, *M*, is determined based on the desired information, P, for specific spectrum states. The training dataset consisting of *K* training samples for DSINet is $D={\left\{\left(x_{i},y_{i}\right)\right\}}_{i=1}^{K}$, where ∀x∈S. The data sample *x* is obtained by the advanced SS system when it senses the spectrum at a specific time, and the sample $x_{i}\in R^{v\times 2}$ represents the I/Q samples as a vector of size v×2. The output $y_{i}=\left(y_{i}^{\left(1\right)},y_{i}^{\left(2\right)},...,y_{i}^{\left(M\right)}\right)$ corresponds to the labels for each of the *M* tasks.

Figure 2 illustrates that DSINet consists of three building blocks: HBs, SBs, and TBs, as we mentioned in the Section 3.3.1. The core aspect of multitask learning is acquiring shared representations from each signal identification task. We explain how each block learns essential shared features below.

**Hard Sharing Blocks** HBs are located at the front of DSINet, close to the input layer. They consist of weight parameters that are shared between all tasks. The HBs capture common features for all tasks, significantly reducing the risk of overfitting for each task. Since all tasks share weight parameters, the use of HBs can minimize the increase in memory usage and inference speed due to the increase in the number of tasks.

There are two feature extraction units: a plain unit and a residual unit. The plain unit consists of two consecutive 1D convolutional (1D Conv) layers followed by a 1D max pooling (1D MaxPool) layer, as illustrated in Figure 3a. The parameters *k*, *f*, and /n represent the kernel size, number of filters, and stride of the 1D Conv layers, respectively.

As the depths of deep neural networks increase, performance improvements are achieved, as has been proven in various fields, such as image processing, natural language processing, and signal processing [37]. However, deeper networks tend to suffer from vanishing gradient problems, leading to slow training. To mitigate this problem, we use the residual unit shown in Figure 3b as the second consideration for feature extraction. In the residual unit, two residual connections are employed to mitigate the effects of vanishing gradients and facilitate faster training processes [37].

**Soft Sharing Blocks** The use of HBs has some advantages in MTL, as mentioned above, but the overall performance may degrade due to competition issues. If the relatedness between tasks is low, each task trains the shared parameters of the HBs to improve its own performance, which results in performance degradations for other tasks. SBs mitigate this task competition problem by exploiting loosely coupled information sharing. In SBs, each task has its own weight parameters to capture independent features per task. In addition, each independent layer shares its information with the other layers by using the cross-stitch unit, as shown in Figure 4.

The cross-stitch unit [36] is a transformation applied between independent layers, and it describes the relationships between different tasks with a linear combination of their activations, which is formulated by:(9)$$\left[\begin{array}{c} o_{l+1}^{\left(T_{1}\right)}\\ \vdots \\ o_{l+1}^{\left(T_{n}\right)} \end{array}\right]=\left[\begin{array}{ccc} \alpha_{11} & \cdots & \alpha_{1n}\\ \vdots & \vdots & \vdots \\ \alpha_{n1} & \cdots & \alpha_{nn} \end{array}\right]\left[\begin{array}{c} o_{l}^{\left(T_{1}\right)}\\ \vdots \\ o_{l}^{\left(T_{n}\right)} \end{array}\right]\;$$
where ${\left\{o_{l+1}^{\left(T_{i}\right)}\right\}}_{i=1}^{n}$ is the transformed output associated with tasks $T_{1}$ to $T_{n}$, $\alpha_{ij}$ is a trainable parameter associated with tasks $T_{i}$ and $T_{j}$, and ${\left\{o_{l}^{\left(T_{i}\right)}\right\}}_{i=1}^{n}$ is the input of the cross-stitch unit. By jointly training the parameters $\alpha_{ij}$, the cross-stitch unit learns the relationship between tasks by itself.

**Task-Specific Blocks** In the MTL settings, the outputs corresponding to each task are highly heterogeneous. For example, the output of the classification task is discrete, while the output of the regression task is continuous. In this case, the cross-stitch unit in the SBs may result in undesired training effects due to the use of a linear combination of features in which one is related to a discrete output, while the other is related to a continuous output. In addition, it may be more difficult to learn features for some tasks than others, which may require more feature extraction layers. To mitigate these problems, we employ TBs, which have independent weight parameters without information-sharing mechanisms, such as cross-stitch units.

## 4. Case Studies

### 4.1. Methodology

In this section, we present three case studies to analyze the performance of DSINet: active primary user signal identification, CTI identification, and signal identification for 5G-advanced and 6G. For each case study, we evaluate the overall performance of DSINet by constructing realistic simulation scenarios that do not assume homogeneous devices, static access methods, or nearly noise-free environments. To ensure the objectivity of DSINet’s evaluations across multiple tasks, we compare its performance with that of other models for each task. The details of case studies are shown in Table 1.

In Case Study 1 (CS-1), we evaluate the performance of DSINet in spectrum sharing scenarios and compare it with the existing shallow signal identification approaches, WTC [40], DeepSense [41], RSC [42]. To ensure a fair comparison, we implement the DSINet variants with input shapes that match those of the compared methods. In Case Study 2 (CS-2), we investigate CTI scenarios by comparing DSINet with a single network (SSINet) that has the same architectural style as that of DSINet but with a single task. In both CS-1 and CS-2, we conduct task ablation studies to investigate the associations between the tasks and how each task affects DSINet. In Case Study 3 (CS-3), we consider complex heterogeneous network scenarios in 6G where both CS-1 and CS-2 co-occur. We study the effects of different DSINet configurations, including the selection of the feature extraction block, the proportions of HBs and SBs, and the addition of TBs. We also analyze the efficiency of DSINet in terms of memory usage and the number of operations. CS-1 and CS-2 focus on comparative evaluations, while CS-3 focuses on a performance evaluation to extract our best DSINet model.

### 4.2. Case Study 1: Active PU Signal Identification

The wireless technologies of primary users considered in CS-1 are LTE (LTE-U), 5G (NR-U), and Wi-Fi (Wireless LAN) operating in the sub-6 GHz frequency band. The fundamental concern in CS-1 is hole detection, $T_{hd}$, and we consider a wideband spectrum as DeepSense [41]. The waveforms of LTE, 5G, and Wi-Fi are generated using MATLAB’s LTE Toolbox, 5G Toolbox, and WLAN Toolbox, respectively. We created 16 holes with approximately 1 MHz bandwidth, which is sufficient for wireless communication technologies, such as Bluetooth, ZigBee, and LoRa, while leaving room for guard bands. In this case, the LTE signals have a 180 kHz resource block size, while the 5G NR-U signals can have a resource block size of either 180 kHz, 360 kHz, or 720 kHz. The Wi-Fi signals, which can be generated based on either the IEEE 802.11ax or IEEE 802.11be standard, consist of a resource unit composed of multiple subcarriers, each with a bandwidth of 78.125 kHz. The comparison models in signal classification have different shapes and even different features, using a 256-size RSSI [40] and a 64 × 64-size STFT image [40]. For each task, the input vector size of the comparison models is different, so we constructed the models as follows for fair comparison.

DSINet-IQ128:For the hole detection task $T_{hd}$, DSINet-IQ128, which is a DSINet variant, is implemented for comparison with DeepSense, which takes an input shape of (128, 2). DSINet-IQ128 has only one SB with the number of filters, kernel size, and stride equal to (32, 3, 2). For the task ablation study, DSINet-IQ128 (all), DSINet-IQ128 (hd + sc), and DSINet-IQ128 (hd + mc) are compared.DSINet-IQ1024:In the modulation classification task, the input of the comparison methods, including RSC-VGGNet and RSC-ResNet, is an IQ vector with a shape of (1024, 2). Therefore, DSINet-IQ1024 with a task combination of (all, mc + sc, mc + hd) is implemented with the same input as that of the comparison targets.DSINet-IQ4096:To conduct a comparison in the signal classification task, we implement DSINet-IQ4096, which has the same input shape as WTC-CNNIQ. DSINet-IQ4096 consists of five HBs and one SB. For the task ablation study, DSINet-IQ4096 variants with task combinations of (all, sc + hd, sc + mc) are implemented.

### 4.3. Case Study 2: CTI Identification

The purpose of models in CS-2 is an identification scheme that enables adaptive CCA to mitigate interference by varying the power levels assigned to different wireless technologies. So, we consider $T_{pe}$, estimating the average power of each signal, as the main task in CS-2. The waveforms of Bluetooth and Zigbee are generated using MATLAB’s Bluetooth Toolbox and Communication Toolbox. The configuration of Wi-Fi signals is based on various WLAN standards, such as IEEE 802.11g (both 5 MHz and 20 MHz), IEEE 802.11n, IEEE 802.11ac, IEEE 802.11ax, and IEEE 802.11be, adding complexity to CTI identification.

In CS-2, DSINet is composed of three HBs, one SB, and one TB(pe). The TB(pe) is only applied for the power estimation task $T_{pe}$ because task $T_{pe}$ deals with a regression problem in which the outputs have different characteristics from those of other classification tasks. For the task ablation study, the four DSINet variants are implemented, which are related to the task combinations of (all, sc + mc, sc + pr, mc + pr). Note that even though $T_{sc}$ is considered a multilabel classification task due to the mix of signals, we assigned a label of $y^{\left(1\right)}$ to multiclass classification for better performance through one-hot labeling.

### 4.4. Case Study 3: Signal Identification for 5G-A/6G

In this study, we leverage DSINet to estimate four tasks: the signals present in the spectrum, the channels in use, the average power utilized by each signal, and the modulations in use. The T for CS-3 involves adding a used channel classification task, $T_{cc}$, to CS-2, which aims to identify PU behaviors for spectrum sharing while identifying CTI. The minimum channel size of wireless technology, Bluetooth, is 1 MHz within 20 MHz bandwidth. In CS-3, we implement a DSINet variant consisting of a block combination of {HB+TB,HB+SB,HB+SB+TB} to evaluate the effects of HBs, SBs, and TBs, which are the core components of DSINet. In addition, we implement and comparatively analyze five DSINet variants with HB:SB proportions of {0:4,1:3,2:2,3:1,4:0} to analyze the effect of this proportion.

DSINet-HB3:TB${}_{all}$1:DSINet-HB3:TB${}_{all}$1 consists of three HBs and one TB${}_{all}$. TB${}_{all}$ indicates that all TBs are connected to each task so that DSINet-HB3:TB${}_{all}$1 has exactly the same setting as traditional MTL approaches (hard parameter sharing) [33].DSINet-HB3:SB1:DSINet-HB3:SB1 is composed of three HBs and one SB. The SB is employed to maximize the information sharing between tasks by using the cross-stitch unit.DSINet-HB3:SB1 + TB${}_{pe}$:DSINet-HB3:SB1 + TB${}_{pe}$ has three HBs and one SB, similar to DSINet-HB3:SB1, but additionally uses a TB${}_{pe}$ connected to the regression task $T_{pe}$.DSINet-HBx:SBy + TB${}_{pe}$ variants for proportion analysis:To analyze the proportion of HB and SB, we implement DSINet-HBx:SBy + TB${}_{pe}$ variants with HB:SB proportions of {0:4,1:3,2:2,3:1,4:0} are utilized.

## 5. Experimental Results and Analysis

### 5.1. Analysis of Case Study 1

#### 5.1.1. Hole Detection Performance

Figure 5 presents the hole detection accuracies of four models: DeepSense [41] and three DSINet models. Our model, DSINet-IQ128(hd+sc), outperforms DeepSense by approximately 3.3% at an SNR of −4 dB, exhibiting the largest accuracy difference. Notably, the section with the most significant performance improvement due to MTL is from an SNR of −8 dB to −2 dB. Conversely, MTL’s effect is relatively insignificant in sections where holes can be easily detected or where detection is almost impossible. All three of our models show promising results compared to DeepSense, which is a single-hole detection model, by sharing task features through MTL.

Interestingly, the model considering $T_{hd}$ + $T_{sc}$ demonstrates better performance than the model with all tasks. Upon closer inspection of Table 2, the accuracy gap between our models is approximately 1% at −4 dB. Similarly, the $F_{1}score$ ranks in descending order are as follows: DSINet-IQ128(hd+sc), DSINet-IQ128(all), and DSINet-IQ128(hd+mc). Our results indicate that $T_{sc}$ has the most significant impact on $T_{hd}$ performance.

#### 5.1.2. Signal Classification Performance

It can be seen that WTC-CNNRSSI and WTC-CNNSTFT have relatively poor results compared not only to those of our models but also to those of WTC-CNNIQ, as shown in Figure 6. Such results were also shown in the single-task dataset in their original paper. WTC-STFT does not perform particularly well, and it is estimated that most information is lost in the process of resizing to a 64 × 64 image. Our three models as well as WTC-CNNIQ all achieve approximately 100% accuracy above −6 dB. The effect of MTL appears below an SNR of −8 dB, which is the starting point of the performance difference between WTC-CNNIQ and our models.

Table 3 shows that all our models perform signal classification with similar performance to each other. Because the task is easy, the performance differences between our models are insignificant, up to 0.4% (we do not consider the case where two or more signals exist simultaneously, so the task is easy to solve).

#### 5.1.3. Modulation Classification Performance

The modulation classification task compares our three models with RSC models [42]. Figure 7 shows that $T_{mc}$ is much more difficult than $T_{sc}$. Even at an SNR of 4 dB, only 90% accuracy is achieved. In addition, the performance of DSINet-1024, especially DSINet-1024(mc+sc), is better than that of the other models in the SNR range from −18 dB to −8 dB. DSINet-1024(mc+sc) exhibits an accuracy difference of 19% at −12 dB from RSC-VGGNet, which is the model with the lowest performance.

What is unusual is that RSC-ResNet, a single modulation model, achieves better performance than our DSINet-1024(all). Sharing information about all tasks actually interrupts the MC performance improvement. In other words, there is competition among tasks. However, it can be said that our model is relatively stable, given that DSINet-1024(all) has lower accuracy than RSC-ResNet but high recall (the correct identification percentage attained for a certain class), as shown in Table 4. The best model is DSINet-1024(mc+sc), which only shares two hole detection tasks, and it is better than RSC-ResNet.

#### 5.1.4. Model Performance in Terms of Execution Time

At the end of CS-1, we examine the model size and FLOPs, as described in Table 5. As emphasized in the DeepSense paper, DeepSense can perform hole detection with a very small model size and high speed. Our model also exhibits high computational performance on the order of three decimal places in GFLOPs. Although the model size is twice as large, increasing the amount of memory is not a major problem in terms of cost and system complexity. The fact that it produces results for three tasks simultaneously with almost the same size and speed can be considered a remarkable performance. Furthermore, it may be even better when only two tasks are considered, e.g., DSINet(hd+sc).

RSSI and STFT are small and fast models but require additional modules, such as a preprocessing module for calculating RSSIs or STFTs. In particular, it can be expected that the inference speed of the system equipped with WTC-STFT might be very slow because I/Q samples must be transformed with the STFT and even converted to images.

### 5.2. Analysis of Case Study 2

In CS-2, since only the path loss and AWGN channels are applied as the average power estimation task; the performance differences between the models are not clearly revealed. Therefore, we analyze the average performance of each model across the entire test dataset (all SNRs).

#### 5.2.1. Task Ablation Study for Signal Classification

As seen in Table 6, all models classify signals well, even when the signals have mixed data samples (the Accuracy, Recall, Precision, and $F_{1}score$ values all exceed 90%). In particular, as in CS-1, DSINet(sc+mc) achieves the highest performance. Since not much noise is contained in the data sample, there are almost no performance differences between the models, but the single model performs slightly better than DSINet(all). However, DSINet(all) estimates two additional tasks with little performance degradation. Even in this case study, the results show that to improve the performance of a task, only task information related to that task should be shared.

#### 5.2.2. Task Ablation Study for Power Estimation

The average power levels estimated by the models are shown in Table 7. It can be seen that almost all models estimate the average power well, around plus or minus 2.75 (the mean absolute error). When the evaluation metric is the $R^{2}score$, DSINet(pe+sc), which achieves the best performance, produces a score of 94.2%. The power estimation yielded by a model with a resolution of 3 dBm is sufficient for performing adaptive CCA. DSINet provides very useful information for developing a CTI mitigation strategy by classifying signals while estimating the power of each signal.

#### 5.2.3. Task Ablation Study for Modulation Classification

Unlike CS-1, in this study, the difficulty of modulation classification is increased by mixing signals. In particular, the recall values of all models are low, as shown in Table 8, which means that the probability of being correct is low. In modulation classification, when signals are mixed simultaneously, the I/Q values are mixed and distorted in the time domain, so a low value is obtained.

We can see that signal classification is not helpful for modulation classification. However, the power values that exist in different spectrum spaces are helpful to some extent, as shown in the performance of DSINet (mc + pe).

In summary, it is confirmed that DSINet achieves better performance than the single-task learning model (when specific tasks are considered) and that it solves many tasks simultaneously with just one model. This means that DSINet can provide abundant spectrum information, which is necessary for establishing a strategy corresponding to each wireless technology in a scenario where CTI frequently occurs.

### 5.3. Analysis of Case Study 3

#### 5.3.1. Residual Unit vs. Plain Unit

DSINet’s performance is significantly affected by how it utilizes the feature extraction units within a block, as with all other AI models. As described in Section 3.3, we consider two feature extraction units, a residual unit and a plain unit, and the performance achieved on each task when these units are applied to DSINet is shown in Figure 8. The scores obtained for the classification tasks, including $T_{sc}$,$T_{cc}$ and $T_{mc}$, represent accuracy, while the scores obtained for the regression task, which is $T_{pe}$, represent $R^{2}score$s. The residual unit achieves good performance in terms of deep signal identification as well as in various fields. The residual unit solves the vanishing problem, and the network depth is relatively deep compared to the plain unit, so it achieves scores of 0.908, 0.975, 0.894, and 0.930 for each task.

#### 5.3.2. Analysis of the Proportions of HBs and SBs

As mentioned in design principle 1, our model has a hybrid shared network structure with HBs and SBs. To investigate how the proportions of HBs and SBs affect the performance of DSINet, we evaluate DSINet-HBx:SBy + TB${}_{pe}$ variants in which the proportions of x and y are set as 0:4, 1:3, 2:2, 3:1, 4:0.

Figure 9 shows the performance of the DSINet-HBx:SBy + TB${}_{pe}$ variants, in which the y-axis indicates the score, and the x-axis is the ratio of HBs to SBs. For classification tasks including $T_{sc}$,$T_{cc}$, and $T_{mc}$, the accuracy metric is used for the scoring function. For $T_{pe}$, which is the regression task, we use the $R^{2}score$ as the scoring function.

As depicted in Figure 9, the DSINet variant with HB4:SB0 achieves the worst performance among all the variants. HB4:SB0 means that all feature extraction layers (weight parameters) are fully shared between all tasks. Therefore, each task cannot learn its own features, leading to the task competition problem, which frequently occurs in heterogeneous multitask learning settings, such as CS-3.

The overall performance of DSINet increases sharply at the HB3:SB1 point on the x-axis, and thereafter, the performance tends to increase slightly as the proportion of SBs increases. Since the SBs allow each task to learn its own features through independent weight parameters, the overall performance is enhanced. However, the use of too many SBs results in high memory usage and a slow inference speed. Figure 10 and Figure 11 show the number of parameters (memory usage) and the number of floating point operations (FLOPs) according to the increase in the SB proportion, respectively. The number of parameters indicates a small memory usage level of approximately 199.4K in SB0, but this usage gradually increases as SB increases, and in SB4, the memory usage of 356.6K exceeds that of four single models (gray box). The number of FLOPs also increases gradually, similar to the number of parameters, and it increases sharply, especially starting from SB2. The numbers of parameters and FLOPs can increase more rapidly as the number of tasks increases. Therefore, we select the DSINet variant with HB3:SB1 as the best model, as it exhibits rapid performance improvement without significantly increasing the numbers of parameters and FLOPs.

#### 5.3.3. Analysis of the Effects of the TBs

According to design principle 2, we employ TBs as additional building blocks for DSINet. In this subsection, DSINet-HB3:SB1 + TB${}_{pe}$, which is the best model selected in the previous subsection, is compared with DSINet-HB3:SB1 and DSINet-HB3:TB${}_{all}$1 to evaluate the effect of the TB on DSINet. DSINet-HB3:SB1 has only HBs and SBs but no TBs. On the other hand, DSINet-HB3:TB${}_{all}$1 consists only of HBs and TBs. TB${}_{all}$ means that all tasks have a specific TB, while TB${}_{pe}$ means that only task $T_{pe}$ has a TB.

On average, as described in Table 9, DSINet-HB3:SB1 + TB${}_{pe}$ outperforms the other models with scores of 0.9084, 0.9752, 0.8935, and 0.9296. Among DSINet-HB3:SB1 and DSINet-HB3:TB${}_{all}$1, DSINet-HB3:TB${}_{all}$1 achieves better performance.

This trend appears similarly in the SNR-based performance comparison. Figure 12 shows the performance of DSINet-HB3:SB1+TB${}_{pe}$, DSINet-HB3:SB1 and DSINet-HB3:TB${}_{all}$1 by SNR and by task. In a high-SNR environment, the performances of all DSINets are similar to each other, but as the SNR decreases, the performance of each model is significantly different. Even at SNRs of −5 or lower, DSINet-HB3:SB1 + TB${}_{pe}$ achieves the highest performance by a large margin. Among the other two models, DSINet-HB3:TB${}_{all}$1 exhibits slightly better performance than DSINet-HB3:SB1, at approximately 4%. This indicates that even if an independent layer is provided for each task through the use of SBs, the performance is not improved without TBs.

Again, CS-3 contains highly heterogeneous tasks, including the classification tasks $T_{sc}$, $T_{cc}$, and $T_{mc}$ and the regression task $T_{pe}$. In a classification task, the output is a discrete value of 0 or 1, whereas in a regression task, the output has a continuous value between 0 and 1. In this case, the cross-stitch unit of the SBs shares information between task layers through the linear transformation of discrete features and continuous features, so the performance between tasks may degrade. On the other hand, if a TB is used for a specific task ($T_{pe}$ in our case), task-specific layers near the output are additionally employed, so the SB features are not significantly affected by the output characteristics. Therefore, DSINet-HB3:SB1 + TB${}_{pe}$ can accommodate the advantages of SBs without compromising performance due to the output characteristics.

#### 5.3.4. Analysis of the Model Size and FLOPs with a Task Size Increase

To see the model size and FLOPs more clearly, we generalize the task to a task with the same one-hot labels and derive the results. In Figure 13, the optimal combination of DSINet solves four tasks with less than 30% of the parameters used by a single model set, for which the model size is 344.1K. In terms of computational performance, our model does not slow down because its FLOPs increase almost insignificantly, unlike the single model set that is slowed by n times every time the task increases, as shown in Figure 14. Although the results are not shown in the figure, our model performs well with a model size of 271.6K vs. 430.2K and FLOPs of 0.014 GFLOPs vs. 0.053 GFLOPs, even when the number of tasks is 5. We believe that it is desirable to apply multitask learning to our DSINet to derive rich spectrum state information while maintaining acceptable model size and speed levels in wireless systems.

## 6. Conclusions

In this paper, we proposed DSINet, a multitask learning-based deep signal identification network for advanced spectrum sensing systems, with a focus on preparing for the upcoming 5G-advanced and 6G systems. We presented three important types of blocks within DSINet, hard sharing blocks, soft sharing blocks, and task-specific blocks, that maximize the effect of sharing characteristics in multitask learning and support the deep identification of signal footprints in multidimensional spectrum spaces.

We evaluated DSINet in three challenging case studies where heterogeneous devices used the spectrum with complex dynamics. In Case Study 1, we compared DSINets with three other model types, including DeepSense, WTCs, and RSCs, in different tasks, such as hole detection, signal classification, and modulation classification. Our results showed that DSINet outperformed these models, with performance improvements of 3.3%, 3.3%, and 5.7%, respectively. In Case Study 2, we conducted task ablation studies to understand the contributions of tasks and found that DSINet provided rich information while exhibiting better performance than single models in all tasks. Through experiments, we discovered that sharing only related tasks is better than sharing all identification tasks unilaterally in terms of performance. The block-based model structure makes it easy to consider this factor in the model design stage. In the last case study, we analyzed DSINets designed with different numbers of HBs, SBs, and TBs under the model design principles to determine the performance differences yielded for each task and derived the best model. The best DSINet achieved 98%, 99%, and 94% accuracy and an $R^{2}score$ of 0.96 in signal classification, channel classification, modulation classification, and power estimation, respectively, even at a relatively low SNR of –4 dB. This performance is significant in practical use, as it came from a model with 225.2 K model parameters and 0.013 GFLOPs (the individual model sets required 344.1 K model parameters and 0.043 GFLOPs).

We expect that DSINet will play a crucial role in supporting seamless spectrum sharing and resource management in the future network environments of 5G-advanced and 6G, where dynamic spectrum sharing and cross-technology interference frequently occur.

## Figures and Tables

**Figure 1 sensors-23-09806-f001:**
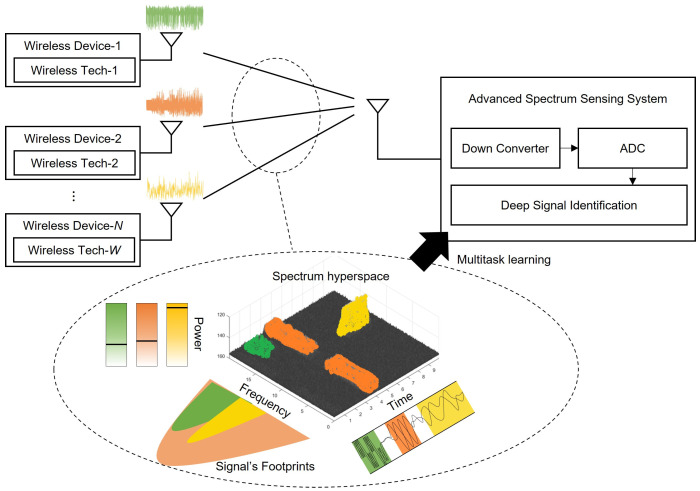
Advanced spectrum sensing scenario in the spectrum hyperspace.

**Figure 2 sensors-23-09806-f002:**
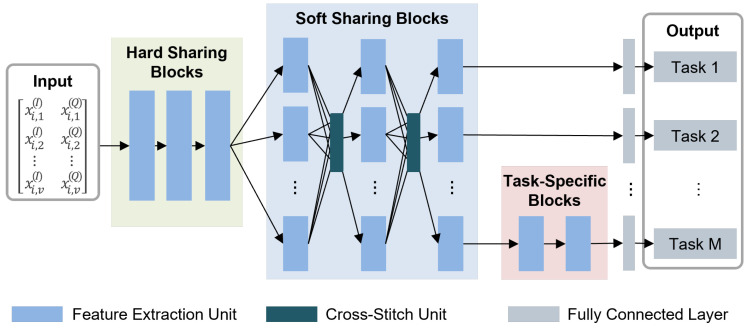
Structure of the DSINet.

**Figure 3 sensors-23-09806-f003:**
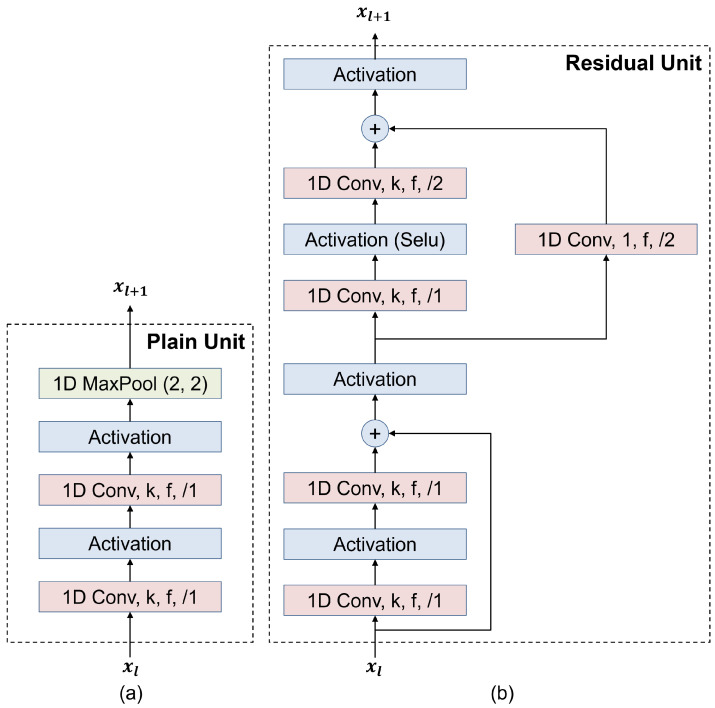
Structure of the (**a**) plain unit and (**b**) residual unit.

**Figure 4 sensors-23-09806-f004:**
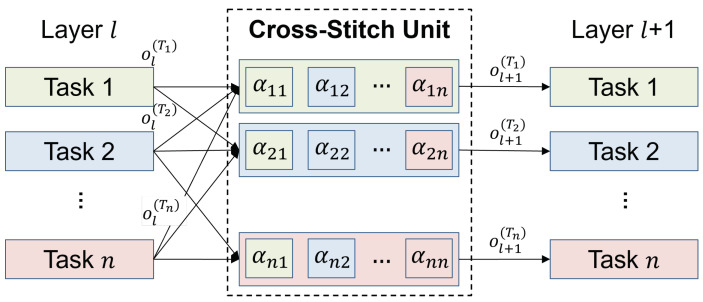
Structure of the cross-stitch unit.

**Figure 5 sensors-23-09806-f005:**
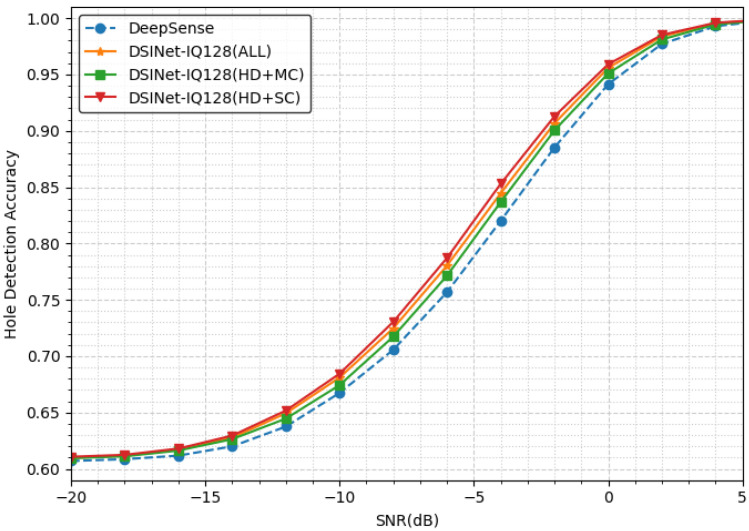
The hole detection accuracies achieved under different SNRs.

**Figure 6 sensors-23-09806-f006:**
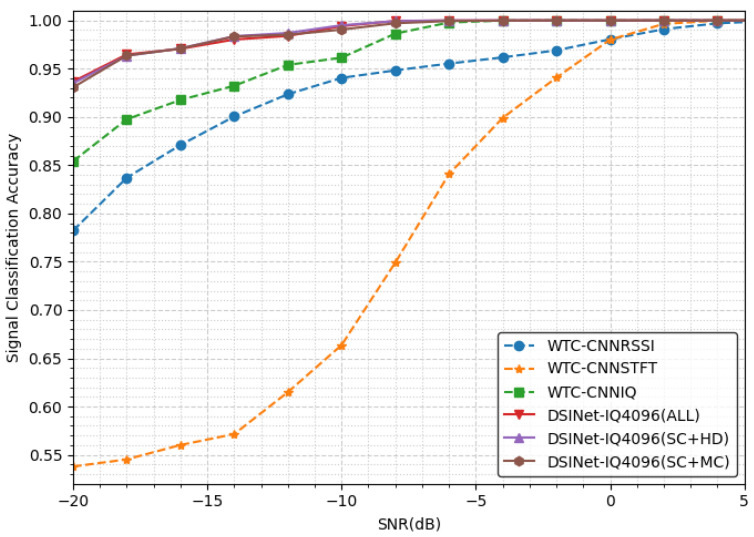
The signal classification accuracies achieved under different SNRs.

**Figure 7 sensors-23-09806-f007:**
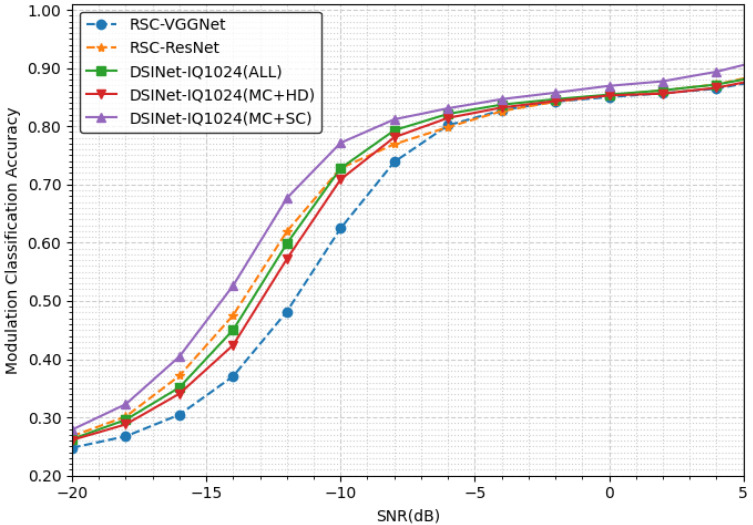
The modulation classification accuracies achieved under different SNRs.

**Figure 8 sensors-23-09806-f008:**
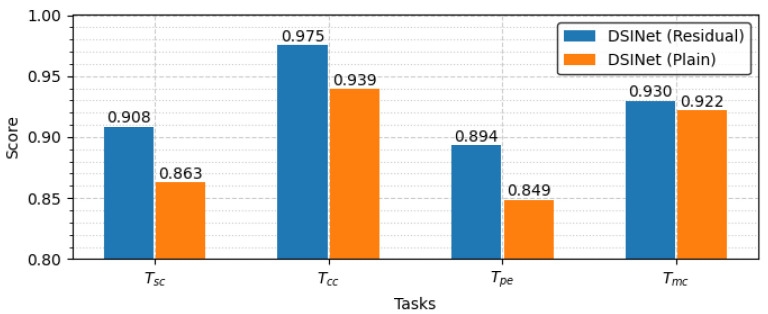
Comparison between the residual unit and plain unit.

**Figure 9 sensors-23-09806-f009:**
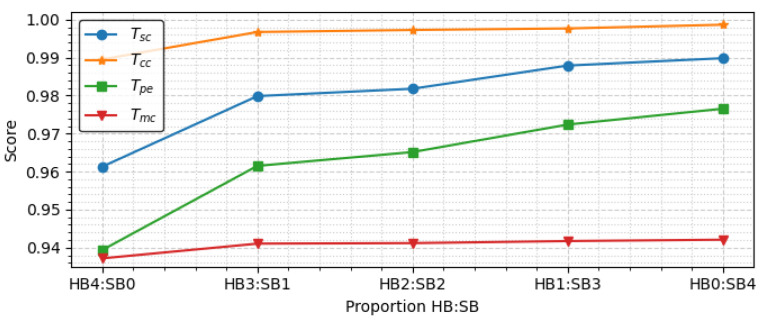
Score comparison according to the proportions of HBs and SBs (SNR: −4 dB).

**Figure 10 sensors-23-09806-f010:**
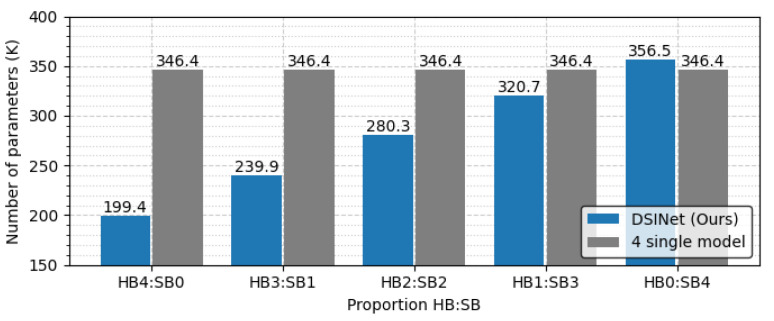
Memory usage comparison according to the proportions of HBs and SBs.

**Figure 11 sensors-23-09806-f011:**
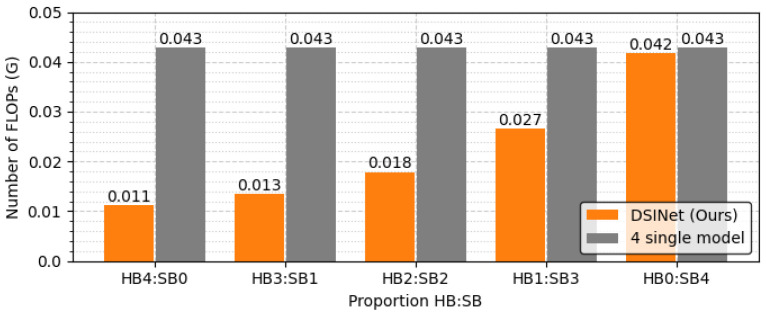
FLOPs comparison according to the proportions of HBs and SBs.

**Figure 12 sensors-23-09806-f012:**
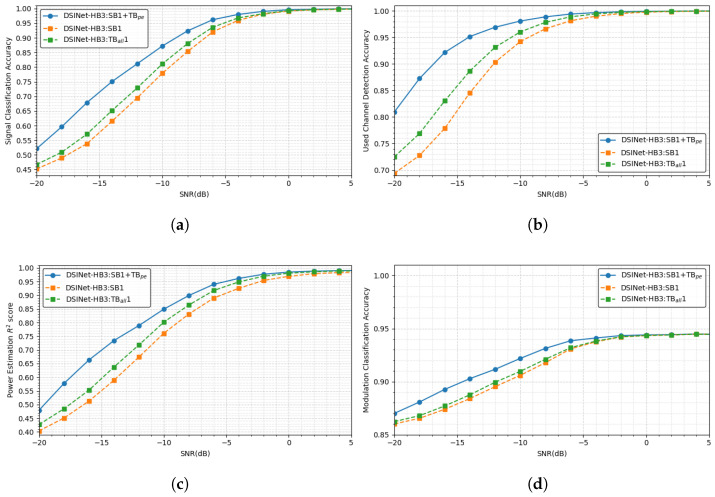
Performance comparison among the DSINet variants according to the block combinations in (**a**) $T_{sc}$, Accuracy of signal classification; (**b**) $T_{cc}$, Accuracy of used channel detection; (**c**) $T_{pe}$, $R^{2}score$ of power estimation; and (**d**) $T_{mc}$, Accuracy of modulation classification.

**Figure 13 sensors-23-09806-f013:**
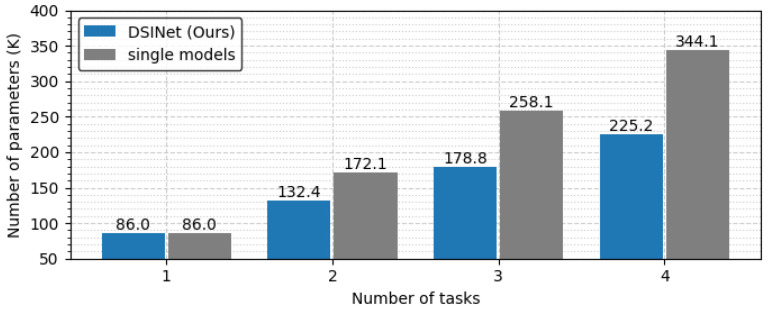
Memory usage comparison between DSINet and the single models.

**Figure 14 sensors-23-09806-f014:**
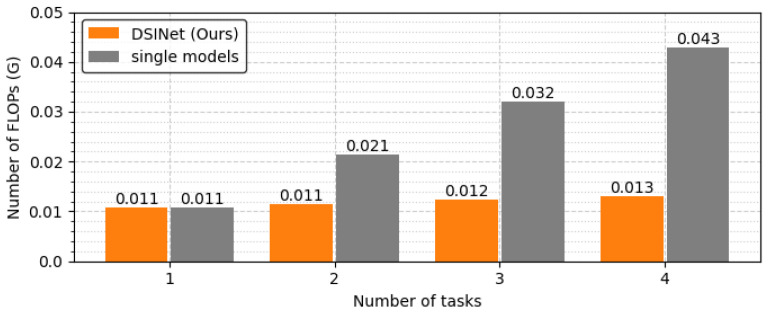
FLOPs comparison between DSINet and the single models.

**Table 1 sensors-23-09806-t001:** Details of Case Studies.

	Case Study 1 (CS-1) Active PU Signal Identification	Case Study 2 (CS-2) CTI Identification	Case Study 3 (CS-3) Signal Identification for 5G-A/6G
Purpose	Primary user in sub-6 GHz	Derive spectrum usage characteristics of Wireless devices in 2.4 GHz unlicensed band	Wireless devices in complex networks
Wireless Tech’	W={LTE,5G,Wi-Fi}	W={Wi-Fi,Bluetooth,Zigbee} Two or more wireless tech’ can coexist	W={Wi-Fi,Bluetooth,Zigbee} Two or more wireless tech’ can coexist
Tasks	$T_{sc}:x\to y^{\left(1\right)}\in {\left\{0,1\right\}}^{\left|W\right|}$ $T_{hd}:x\to y^{\left(2\right)}\in {\left\{0,1\right\}}^{\left|hd\right|}$ $T_{mc}:x\to y^{\left(3\right)}\in {\left\{0,1\right\}}^{\left|mc\right|}$	$T_{sc}:x\to y^{\left(1\right)}\in {\left\{0,1\right\}}^{\left|W\right|}$ $T_{pe}:x\to y^{\left(2\right)}\in {\left[0,1\right]}^{\left|W\right|}$ $T_{mc}:x\to y^{\left(3\right)}\in {\left\{0,1\right\}}^{\left|mc\right|}$	$T_{sc}:x\to y^{\left(1\right)}\in {\left\{0,1\right\}}^{\left|W\right|}$ $T_{cc}:x\to y^{\left(2\right)}\in {\left\{0,1\right\}}^{\left|cc\right|}$ $T_{pe}:x\to y^{\left(3\right)}\in {\left[0,1\right]}^{\left|W\right|}$ $T_{mc}:x\to y^{\left(4\right)}\in {\left\{0,1\right\}}^{\left|mc\right|}$
Data Gen’	K=6,881,175,eachwithanSNR∈[−20,20] Training/validation/test set are randomly sampled at 80%, 10%, and 10% ratios sc=W |hd|=16 $\begin{array}{c} mc=\{\mathrm{BPSK},\mathrm{QPSK},16\mathrm{QAM},64\mathrm{QAM},\\ 256\mathrm{QAM},1024\mathrm{QAM},4096\mathrm{QAM}\} \end{array}$ x∈{IQ,RSSI,STFT}, (4096,2), (128,2), and (1024,2) I/Q vectors are generated for each task, (256,1) RSSI sample and (64,64) STFT image are generated for $T_{sc}$ $y_{i,c}^{\left(1\right)}\in \left\{0,1\right\}\;\mathrm{for}\;c\in sc,\sum_{c\in sc}y_{i,c}^{\left(1\right)}=1,\forall i$ $y_{i,c}^{\left(2\right)}\in \left\{0,1\right\}\;\mathrm{for}\;c\in hd,\sum_{c\in hd}y_{i,c}^{\left(2\right)}\leq \left|hd\right|,\forall i$ $y_{i,c}^{\left(3\right)}\in \left\{0,1\right\}\;\mathrm{for}\;c\in mc,\sum_{c\in mc}y_{i,c}^{\left(3\right)}=1,\forall i$	K=6,300,000,eachwithanSNR∈[−20,20] Training/validation/test set are randomly sampled at 80%, 10%, and 10% ratios $\begin{array}{cc} sc= & \{\mathrm{Wi}-\mathrm{Fi},\mathrm{BT},\mathrm{ZB},\mathrm{Wi}-\mathrm{Fi}\;+\;\mathrm{BT},\mathrm{Wi}-\mathrm{Fi}+\mathrm{ZB},\\ & \mathrm{ZB}\;+\;\mathrm{BT},\mathrm{Wi}-\mathrm{Fi}\;+\;\mathrm{BT}\;+\;\mathrm{ZB}\} \end{array}$ $\begin{array}{c} mc=\{\mathrm{GFSK},\mathrm{BPSK},\mathrm{QPSK},\mathrm{OQPSK},16\mathrm{QAM},\\ 64\mathrm{QAM},256\mathrm{QAM},1024\mathrm{QAM},4096\mathrm{QAM}\} \end{array}$ x∈IQ, (1024,2) I/Q vectors are generated $y_{i,c}^{\left(1\right)}\in \left\{0,1\right\}\;\mathrm{for}\;c\in sc,\sum_{c\in sc}y_{i,c}^{\left(1\right)}=1,\forall i$ $y_{i,c}^{\left(2\right)}\in \left[0,1\right]\;\mathrm{for}\;c\in W,\forall i$ $y_{i,c}^{\left(3\right)}\in \left\{0,1\right\}\;\mathrm{for}\;c\in mc,\sum_{c\in mc}y_{i,c}^{\left(3\right)}\leq \left|W\right|,\forall i$	K=6,300,000,eachwithanSNR∈[−20,20] Training/validation/test set are randomly sampled at 80%, 10%, and 10% ratios $\begin{array}{cc} sc= & \{\mathrm{Wi}-\mathrm{Fi},\mathrm{BT},\mathrm{ZB},\mathrm{Wi}-\mathrm{Fi}+\mathrm{BT},\mathrm{Wi}-\mathrm{Fi}\;+\;\mathrm{ZB},\\ & \mathrm{ZB}\;+\;\mathrm{BT},\mathrm{Wi}-\mathrm{Fi}\;+\;\mathrm{BT}\;+\;\mathrm{ZB}\} \end{array}$ |cc|=20 $\begin{array}{c} mc=\{\mathrm{GFSK},\mathrm{BPSK},\mathrm{QPSK},\mathrm{OQPSK},16\mathrm{QAM},\\ 64\mathrm{QAM},256\mathrm{QAM},1024\mathrm{QAM},4096\mathrm{QAM}\} \end{array}$ x∈IQ, (1024,2) I/Q vectors are generated $y_{i,c}^{\left(1\right)}\in \left\{0,1\right\}\;\mathrm{for}\;c\in sc,\sum_{c\in sc}y_{i,c}^{\left(1\right)}=1,\forall i$ $y_{i,c}^{\left(2\right)}\in \left\{0,1\right\}\;\mathrm{for}\;c\in cc,\sum_{c\in cc}y_{i,c}^{\left(2\right)}\leq \left|cc\right|,\forall i$ $y_{i,c}^{\left(3\right)}\in \left[0,1\right]\;\mathrm{for}\;c\in W,\forall i$ $y_{i,c}^{\left(4\right)}\in \left\{0,1\right\}\;\mathrm{for}\;c\in mc,\sum_{c\in mc}y_{i,c}^{\left(4\right)}\leq \left|W\right|,\forall i$
	Sense 20 MHz wideband spectrum Hole size: 1 MHz AWGN and Rayleigh fading	Sense 20 MHz wideband spectrum AWGN Pathloss model [38] with distance from 5 to 20	Sense 20 MHz wideband spectrum Min channel BW: 1 MHz AWGN and Rayleigh fading Pathloss model [38] with distance from 5 to 20
Models	$T_{sc}:$ WTC-{CNNIQ, CNNRSSI, CNNSTFT} $T_{hd}:$ DeepSense $T_{mc}:$ RSC-{VGGNet, ResNet}	SSINet (sc) SSINet (mc) SSINet (pe)	* Comparison of DSINet’s block configuration DSINet-HB3:TB${}_{all}$1 DSINet-HB3:SB1 DSINet-HB3:SB1 + TB${}_{pe}$ DSINet-HBx:SBy + TB${}_{pe}$
	$T_{sc}:$ DSINet-IQ4096 $T_{hd}:$ DSINet-IQ128 $T_{mc}:$ DSINet-IQ1024	* Task ablation study DSINet(all) DSINet(sc+pe) DSINet(sc+mc) DSINet(mc+pe)	

* Details of parameter settings. Adam optimizer ($\beta_{1}=0.9,\;\beta_{2}=0.999,\epsilon =10^{\left(-7\right)})$ with a learning rate of 0.0005. Alpha dropout [39] in fully connected (fc) layer. Scaled exponential linear unit (SELU) [39] activation function is used to activate the functions of all hidden layers, including the 1D convolution, HB, SB, and fully connected layers.

**Table 2 sensors-23-09806-t002:** Performance comparison between DSINet-IQ128 and DeepSense (SNR: −4 dB).

Methods	Accuracy	Recall	Precision	F${}_{1}$ score
DeepSense	0.8202	0.8161	0.8767	0.8452
DSINet-IQ128(all)	0.8447	0.8459	0.8906	0.8675
DSINet-IQ128(hd+mc)	0.8369	0.8264	**0.8947**	0.8590
DSINet-IQ128(hd+sc)	**0.8531**	**0.8601**	0.8919	**0.8756**

**Table 3 sensors-23-09806-t003:** Performance comparison between DSINet-IQ4096 and WTC (SNR: −6 dB).

Methods	Accuracy	Recall	Precision	F${}_{1}$ score
WTC-CNNIQ	0.9539	0.9309	0.9314	0.9265
WTC-CNNRSSI	0.9235	0.8811	0.8964	0.8769
WTC-CNNSTFT	0.6149	0.4200	0.5093	0.4199
DSINet-IQ4096(all)	0.9839	0.9769	0.9741	0.9750
DSINet-IQ4096(sc+hd)	**0.9870**	0.9801	**0.9791**	0.9791
DSINet-IQ4096(sc+mc)	0.9851	**0.9832**	0.9757	**0.9793**

**Table 4 sensors-23-09806-t004:** Performance comparison between DSINet-1024 and RSC (SNR: −12 dB).

Methods	Accuracy	Recall	Precision	F${}_{1}$ score
RSC-VGGNet	0.4818	0.3944	0.5022	0.4719
RSC-ResNet	0.6194	0.4949	0.5968	0.5149
DSINet-1024(all)	0.5986	0.5395	0.5827	0.5501
DSINet-1024(mc+hd)	0.5721	0.5074	0.5492	0.5181
DSINet-1024(mc+sc)	**0.6765**	**0.6481**	**0.6487**	**0.6454**

**Table 5 sensors-23-09806-t005:** Comparison between the model sizes and FLOPs of DSINet and DeepSense, RSCs, WTCs.

Methods	Model Size (K)	FLOPs (G)
DeepSense	65.040	0.00119
RSC-VGGNet	159.431	0.0262
RSC-ResNet	166.791	0.053
WTC-IQ	212.935	0.113
WTC-RSSI	43.747	0.00175
WTC-STFT	55.430	0.0184
DSINet-IQ128	136.796	0.00216
DSINet-IQ1024	180.284	0.0123
DSINet-IQ4096	207.228	0.0464

**Table 6 sensors-23-09806-t006:** Signal classification performance comparison according to a task ablation study.

Methods	Accuracy	Recall	Precision	F${}_{1}$ score
SSINet(sc)	0.9430	0.9228	0.9227	0.9222
DSINet(all)	0.9422	0.9218	0.9219	0.9215
DSINet(sc+pe)	0.9446	0.9252	0.9248	0.9245
DSINet(sc+mc)	**0.9454**	**0.9260**	**0.9259**	**0.9257**

**Table 7 sensors-23-09806-t007:** Power estimation performance comparison according to a task ablation study.

Methods	Mean Absolute Error	$R^{2}\mathrm{score}$
SSINet(pe)	2.7512	0.9356
DSINet(all)	**2.7495**	**0.9407**
DSINet(pe+sc)	**2.7509**	**0.9419**
DSINet(pe+mc)	2.7646	0.9374

**Table 8 sensors-23-09806-t008:** Modulation classification performance comparison according to a task ablation study.

Methods	Accuracy	Recall	Precision	F${}_{1}$ score
SSINet (mc)	0.9377	0.5530	0.6994	0.5693
DSINet (all)	0.9367	0.5344	0.6620	0.5461
DSINet (mc + sc)	0.9369	0.5324	0.6616	0.5428
DSINet (mc + pe)	**0.9377**	**0.5554**	**0.7097**	**0.5711**

**Table 9 sensors-23-09806-t009:** Performance comparison according to the block combinations of DSINet.

Methods	$T_{\mathrm{sc}}$ AccScore	$T_{\mathrm{cc}}$ AccScore	$T_{\mathrm{pe}}$ $R^{2}\mathrm{Score}$	$T_{\mathrm{mc}}$ AccScore
DSINet-HB3:TB${}_{all}$1	0.8804	0.9551	0.8668	0.9250
DSINet-HB3 + SB1	0.8701	0.9452	0.8445	0.9238
DSINet-HB3+SB1 + TB${}_{pe}$	**0.9084**	**0.9752**	**0.8935**	**0.9296**

## Data Availability

Data are contained within the article.

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
