# Peer review of "Multitask Learning-Based Deep Signal Identification for Advanced Spectrum Sensing"

_sensors, 2023, doi:10.3390/s23249806_

Round 1

Reviewer 1 Report

Comments and Suggestions for Authors

In this paper, the authors propose a multitask learning-based deep signal identification network for advanced spectrum sensing systems. The network involves not only classifying signals but also deriving the spectrum usage characteristics of signals across various spectrum dimensions including time, frequency, power, and code. Numerical results are presented and compared against existing shallow signal identification models. I think this work important and interesting. 

Comments on the Quality of English Language

1. row 232: "x"-->"x"

2. Eq.(5):  absolute value of M. What do you mean?

Author Response

Dear Reviewer

 We appreciate the opportunity to further update our submitted paper titled 'Multitask Learning-Based Deep Signal Identification for Advanced Spectrum Sensing'. Thanks to the valuable time and effort of the reviewer, we were able to improve the quality of the manuscript. As the reviewer noted, we fixed some small issues. In particular, your insightful comments helped us identify and correct mis-typed mathematical symbols and equations. Due to your careful review, we also added a beta symbol in Eq. (5) that was previously omitted due to a LaTeX syntax error. Please see the attachment. The revised parts have been highlighted in yellow.

  1. Point-by-point response to Comments and Suggestions for Authors

Comment 1: In this paper, the authors propose a multitask learning-based deep signal identification network for advanced spectrum sensing systems. The network involves not only classifying signals but also deriving the spectrum usage characteristics of signals across various spectrum dimensions including time, frequency, power, and code. Numerical results are presented and compared against existing shallow signal identification models. I think this work important and interesting.;

  • Response: Thank you for your encouraging comments on our paper. We are pleased that you find our work on multitask learning-based deep signal identification for advanced spectrum sensing both important and interesting.
  1. Response to Comments on the Quality of English Language

Point 1: row 232: "x"-->"x";

  • Response: Thank you for pointing out the mis-typed mathematical symbols. This has allowed me to correct them in rows 232, 227, and 229.

Point 2: Eq.(5): absolute value of M. What do you mean?;

  • Response: Since M represents the number of tasks, we initially took its absolute value. However, given that the value of M cannot be negative, it appears unnecessary to display the absolute value separately. We hope that our understanding aligns with the intentions you have pointed out.

Reviewer 2 Report

Comments and Suggestions for Authors

1. The paper proposes a novel multi-task ML-based SS algorithm named DSINet. The proposed algorithm is presented in detail and its performance is thoroughly compared with other ML-based SS algorithms. Extensive tests were run to prove the performance increase of the proposed algorithm over existing algorithms.

2. Define terms denoted by y in (6)-(8).

3. The authors should provide more details regarding the signals used for the training sets of all algorithms, in all scenarios.

Comments on the Quality of English Language

There are some formatting and punctuation errors; some examples: 

- „of the identified signal such as the channel” must be corrected as: „of the identified signal, such as the channel”; 

- „wireless technologies such as WiFi, ZigBee, and” must be corrected as: „wireless technologies, such as WiFi, ZigBee, and”; 

!! do not expand the acronyms after their first definition; example:

- „Signal identification is one of the technologies that can advance spectrum sensing (SS) to function” must be corrected as: „Signal identification is one of the technologies that can advance SS to function”; 

- „user waveforms such as air interface parameters” must be corrected as: „user waveforms, such as air interface parameters”; 

!! Check and correct punctuation, all across the paper

- „refers to the recognize something and validation” must be corrected as: „refers to recognizing something and validation”; 

- „The details of case studies as shown in Table 1.” must be corrected as: „The details of case studies are shown in Table 1.”; 

- „ On average, as described in Table , DSINet-HB3:SB1+TBpe outperforms” must be corrected as: „ On average, as described in Table 9, DSINet-HB3:SB1+TBpe outperforms”; 

!! Etc., etc. 

Author Response

Dear Reviewer

 We appreciate the opportunity to further update our submitted paper titled 'Multitask Learning-Based Deep Signal Identification for Advanced Spectrum Sensing'. Thanks to the valuable time and effort of the reviewer, we were able to enhance the quality of the manuscript. As the reviewer suggested, we added missing terms to the equations and detailed information about the signals included in the datasets. We have also made general corrections to the formatting and punctuation errors that were kindly pointed out. Please see the attachment. The revised parts have been highlighted in yellow.

  1. Point-by-point response to Comments and Suggestions for Authors

Comment 1: The paper proposes a novel multi-task ML-based SS algorithm named DSINet. The proposed algorithm is presented in detail and its performance is thoroughly compared with other ML-based SS algorithms. Extensive tests were run to prove the performance increase of the proposed algorithm over existing algorithms.;

  • Response: Thank you for acknowledging the depth and thoroughness of our multi-task ML-based SS algorithm, DSINet. We are pleased to hear that our detailed presentation and extensive comparative tests have effectively demonstrated the enhanced performance of DSINet over existing algorithms.

Comment 2: Define terms denoted by y in (6)-(8);

  • Response: Thank you for your important comments. As you rightly pointed out, we should have explicitly defined the term 'y' from Eq. (6)-(8). We have now added the definition of 'y' in our paper as you can see in the attached file. We have highlighted the revised parts in yellow.

Comment 3: The authors should provide more details regarding the signals used for the training sets of all algorithms, in all scenarios.;

  • Response: Thank you for pointing this out. We agree that the comment you highlighted is very important from the perspective of paper reproducibility. Therefore, we have added detailed information about the signals included in our dataset in the 'Data Generation' entry of Table 1. Additionally, we have provided detailed descriptions of the considerations made in generating the signals for each case study.
  1. Response to Comments on the Quality of English Language

Point 1: There are some formatting and punctuation errors;

  • Response: I would like to thank you for thoroughly reviewing the paper and identifying formatting and punctuation errors. In addition to the points you highlighted, I have reviewed the entire paper to find and correct any other areas needing improvement.

Reviewer 3 Report

Comments and Suggestions for Authors

This paper presents an approach to deep signal identification network based on multitask learning, for advanced spectrum sensing.

The paper is well written and structured. Results presented in the experimental section of the paper come to support the theoretical background presented in the previous sections.

The paper can be accepted as is.

Author Response

Dear Reviewer

We would like to express our sincere gratitude for your positive and encouraging feedback on our manuscript titled "Multitask Learning-Based Deep Signal Identification for Advanced Spectrum Sensing”. We are pleased to hear that the paper is well-received and that its structure, writing, and experimental results meet your approval. We are honored by your recommendation to accept the paper as it is. Your acknowledgment of our efforts motivates us to continue our research and strive for excellence in our future work. Thank you once again for your constructive and supportive review.